# Transcatheter Closure of Perimembranous Ventricular Septal Defects Including Multifenestrated and Gerbode-Type Defects Using the Lifetech Konar Device

**DOI:** 10.3390/jcm12196370

**Published:** 2023-10-05

**Authors:** Francois Godart, Jean Benoit Baudelet, Alexandre Delarue, Anne Sophie Polge, Olivia Domanski, Said Bichali, Ali Houeijeh

**Affiliations:** 1Department of Pediatric Cardiology and Congenital Heart Disease, Institut Cœur Poumon, CHRU Lille, 59000 Lille, France; jeanbenoit.baudelet@chru-lille.fr (J.B.B.); alexandre.delarue@chru-lille.fr (A.D.); olivia.domanski@chru-lille.fr (O.D.); sail.bichali@chru-lille.fr (S.B.); ali.houeijeh@chru-lille.fr (A.H.); 2Department of Echocardiography and Physiology, Institut Cœur Poumon, CHRU Lille, 59000 Lille, France; annesophie.polge@chru-lille.fr

**Keywords:** pediatric cardiology, congenital heart disease, perimembranous VSD, Konar-MFO device

## Abstract

(1) Transcatheter closure of perimembranous ventricular septal defects (PmVSD) is becoming more attractive and effective with the development of new occluders. The aim of this study was to report a single-center experience in PmVSD closure using the Lifetech Konar-multifunctional occluder (MFO). (2) From March 2019 to October 2022, 43 consecutive patients were enrolled in the study. Among them, 13 had multifenestrated PmVSD including 5 Gerbode-type defects. (3) There were 23 males/20 females, and the median age was 17 years (range 2–68 years). Trivial aortic regurgitation was noticed in 19 patients. Implantation was successful in all patients under general anesthesia. A retrograde approach was used in 35 patients (81%). The retrograde approach was associated with a lower radiation dose (*p* = 0.042) and shorter fluoroscopy time (*p* = 0.002) compared to the antegrade approach. Full occlusion was observed immediately in 12 patients (28%) and in 33 patients (77%) at a median follow-up of 11 months. There were no complications such as embolization, complete atrioventricular block, device dislocation, new onset above grade I, or progression of tricuspid or aortic valve regurgitation. Seven of the thirteen patients with a multifenestrated defect had no residual shunt. The persistent shunts were all trivial intra-prosthetic leaks. (4) MFO is effective and safe for PmVSD closure including multifenestrated/Gerbode-type defects with no complication. However, a longer follow-up remains warranted to establish the safety of this technique.

## 1. Introduction 

Perimembranous ventricular septal defect (PmVSD) is one of the most frequent types of congenital heart disease, which is classically repaired using open-heart surgery. The highly heterogeneous anatomical morphology, the proximity to the aortic and tricuspid valves, the prolapse of the right coronary cusp, and the potential for complete heart block make transcatheter PmVSD closure a complex and challenging procedure. The transcatheter approach has been made possible by the advances in interventional cardiac catheterization and the development of new occluders. Many devices have been used off-label or have been specifically designed for this indication with varying success [1,2,3,4,5]. The purpose of this work is to report a single-center experience using the new Konar-multifunctional occluder (MFO) (Lifetech, Shenzhen, China) for PmVSD closure including for multifenestrated and Gerbode-type defects. 

## 2. Methods

### 2.1. Study Population

From March 2019 to October 2022, all consecutive patients referred for transcatheter PmVSD closure were included in this retrospective single-center study. Indications for closure were defect-associated heart failure symptoms or evidence of significant left-to-right shunt, including left ventricle enlargement on transthoracic echocardiography, a cardiothoracic ratio > 0.5 or pulmonary-to-systemic flow ratio > 1.5, growth failure, recurrent respiratory tract infections, associated aortic insufficiency not above grade II, and a history of endocarditis. Exclusion criteria included PmVSD with aortic prolapse and aortic regurgitation above grade II, significant sub-aortic or infundibular stenosis, severe pulmonary hypertension, a subaortic rim < 2 mm in nonaneurysmal defects, active bacterial infection, body weight < 7 kg, right-sided defect size > 10 mm, and the presence of any associated congenital heart disease requiring cardiac surgery. Before intervention, all patients underwent detailed transthoracic echocardiography (TTE) with VSD size measurement on the left ventricle (LV) and the right ventricle (RV) sides using the parasternal short-axis and long-axis views. Informed consent was signed before the procedure by all patients or their legal guardians. All procedures contributing to this work comply with the ethical standards of relevant national guidelines on human experimentation and with the Helsinki Declaration of 1975, as revised in 2008. This study was registered with CNIL (Commission national de l’informatique et des libertés, N° 1509). Moreover, in the present study, only one patient was also included in the Francisco observational cohort study.

### 2.2. Device

The MF Konar system is a self-expandable double disc made of a nitinol wire mesh developed to close defects from 2 to 10 mm. It has a hybrid design combining a cone-shaped plug similar to the device used for arterial duct closure with a retention disc on the left side and a right-sided disc joined by a soft articulation. Two connecting micro-screw systems are attached on the outer surfaces of both the right and left discs allowing delivery from either the RV or the LV, or both. Device sizes are categorized according to the diameter of the base (D2) and the top (D1) of the conical left disc (Figure 1). This device is available in 8 sizes with the need for a delivery sheath ranging from 4 to 7 F. The 4 larger models are sewn with polytetrafluoroethylene membranes using Nylon threads but the 4 smaller occluders are not.

### 2.3. Procedure

The procedure was performed under general anesthesia using transesophageal echocardiography (TEE) control. A left ventriculography was performed to assess the anatomy of the defect. The choice of device was always determined according to the TEE dimension of the defect on the right ventricular side with possible modifications related to the dimension on the left ventricular side [5]. Device implantation was performed using either a retrograde or antegrade approach. In the retrograde approach, no complete arteriovenous loop was required. Regardless of the approach used, TEE was always used to check after deployment but before release, the device position, the amount of residual shunting, and the presence of aortic or tricuspid valve regurgitation. All patients had 24-h continuous/telemetry monitoring after implantation. The patients were discharged from hospital on the following day, after TTE and electrocardiogram (ECG) control, most often without any treatment or sometimes, at the beginning of our experience, under aspirin (3–5 mg/kg once daily for 6 months). In the case of a residual shunt, hemolysis was determined using blood and urine analyses. Endocarditis prophylaxis was recommended for the first 6 months. Routine outpatient follow-up visits were planned at 1 month, 3 to 6 months, and 12 months after implantation, and thereafter annually including TTE and ECG monitoring. On TTE, the residual shunt was graded according to the color jet width and defined as small (1–2 mm), moderate (2–4 mm), or large (>4 mm) [2,5].

### 2.4. Statistical Analysis

All distributions were tested for normality using the Shapiro–Wilk normality test. Data were shown as mean (M) ± standard deviation (SD) in case of normality, and median (Med) ± interquartile range (IQR) otherwise. For continuous data, an unpaired *t*-test was used in the case of a normal distribution of both tested variables, a paired *t*-test was used for paired variables, and a Wilcoxon–Mann–Whitney test was used otherwise. The alpha risk was set at 5%, and the beta risk was set at 20%. A comparison was conducted between fluoroscopy time and radiation dose, and the antegrade and retrograde approaches with exclusion of other interventional procedures associated with VSD closure.

## 3. Results

### 3.1. Study Population

Forty-three consecutive patients were enrolled in this study including 23 males and 20 females. There were 22 children (age < 18 years) (51%) and 21 adults (49%). The median age at implantation was 17 years (ranging from 2 to 68 years) and the mean weight was 58 ± 27 kg. Three patients had a past history of tricuspid endocarditis, one or two years before implantation, and all refused surgery. One patient had permanent atrial fibrillation. All patients but two had associated VSD aneurysm. The LV was enlarged in end diastole: +1.71 ± 1.30 Z-score. In patients with an enlarged LV but Z-score value under +2, the indications for closure were as follows: exertional dyspnea expressed by the patient (n = 9), presence of associated aortic regurgitation (n = 6), heart failure requiring medication (n = 4), history of endocarditis (n = 2), and need for shunt closure before military enrollment (n = 3). In the TTE long-axis view, the mean VSD diameter on the LV side was 7.67 ± 5.00 mm, and the mean diameter on the RV side was 4.03 ± 1.04 mm. In the TTE short-axis view, the mean VSD diameter on the LV side was 6.43 ± 2.37 mm, and it was 4.03 ± 1.21 mm on the RV side. In 13 patients (30%), there were multiple exits on the right side—the so-called multifenestrated defect—on TTE performed before cardiac catheterization or TEE performed during the procedure. The multifenestrated defect usually had two exits except one that had three exits. In addition, five of the multifenestrated defects were also Gerbode-type VSDs (infravalvular shunt combined with the multifenestrated defect). Moreover, 20 patients (46%) had a small tricuspid regurgitation before implantation (grade I in 18 patients and grade II in 2 patients). Similarly, 19 patients (44%) had a small aortic regurgitation (grade I in 17 patients and grade II in 2 patients); this insufficiency was associated with a mild aortic cusp prolapse in 4 of them. Lastly, three patients had a mild infundibular hypertrophy associated with PmVSD, and one of them had a mild subaortic membrane.

### 3.2. Implantation

Implantation was successful in all patients, but the route for delivery was changed in seven patients from the antegrade to the retrograde approach in five patients, and the retrograde to the antegrade approach in two patients. Overall, implantation was performed using the retrograde approach in 35 patients (81%) and using the antegrade approach in 8 patients (19%). No difference was observed between the antegrade or retrograde approaches regarding age, weight, LV enlargement, or defect size on the right side (*p* = NS). In 29 patients, the release of the device was successful at the first attempt. Multiple attempts were necessary because of incorrect initial device positioning in 14 patients, ranging from two to five attempts. The sizes of implanted devices were as follows: 5–3 mm (n = 1), 7–5 mm (n = 7), 8–6 mm (n = 5), 9–7 mm (n = 6), 10–8 mm (n = 12), 12–10 mm (n = 7), and 14–12 mm (n = 5). Most of the patients received only one device. In two of them with multiple right-sided exits, the persistence of a residual shunt after implantation of a 10–8 mm and 12–14 mm Konar-MFO device yielded to implantation of a vascular plug II (Abbott) with a good final result. Lastly, in another patient, a 7–5 mm device was changed for an 8–6 mm device because the defect was judged a bit larger on echocardiography. In seven patients measuring more than 180 cm, the delivery sheath of the Konar-MFO device was too short for the retrograde approach and was replaced by a Flexor^R^ Check-Flo Performer introducer (length 85 cm) or Flexor^R^ Tuohy-Borst Side-arm introducer (length 90 cm) from Cook medical (Bloomington, IN).

For the 13 patients with multifenestrated defects (including the 5 with Gerbode-type VSDs), defect closure was always performed using a retrograde approach. The guide wire was advanced within the defect from the left ventricle to the right ventricle in an attempt to cross the largest defect. The device size D1 was chosen 2 mm greater than the largest diameter of the right orifice (RV size) as usual. Of more importance, the device had to be placed inside the aneurysm and the device size D2 was at least 1 or 2 mm larger than the diameter of the left-sided orifice to cover all the aneurysm and different holes and to close the LV entry. In case of immediate periprosthetic residual shunt, another device could be inserted as described above.

The median fluoroscopy time was 7.09 min, IQR 7.58, and the median radiation dose was 7.24 Gy·cm^2^, IQR 11. In the comparative antegrade/retrograde route study, the retrograde approach was associated with a shorter fluoroscopy time (median 6.13 min, IQR 5.64 min versus 19.2 min, IQR 8.93, *p* = 0.002) and a lower radiation dose (6.72 Gy·cm^2^, IQR 9.12 versus 18.3 Gy·cm^2^, IQR 19.3, *p* = 0.042) compared to the antegrade approach.

At implantation, the immediate TEE after release showed full occlusion in 12 patients (28%) and the remaining 31 patients had a small residual shunt (72%). One patient presented with transient Mobitz I-type atrioventricular (AV) block at release. On ECG, the PR interval changed from 170 ± 26 ms before implantation to 174 ± 54 ms after implantation (*p* = 0.65). When the PR interval was indexed to the RR interval, no change was noticed in PR/$\sqrt{RR}$ before and after implantation (*p* = 0.65). The QRS axis moved from 42 ± 27° to 51 ± 50° (*p* = 0.25). No arrhythmia was noticed on telemetry monitoring following implantation. All patients were discharged one day after device release. Only the first four patients received aspirin after implantation during the early experience, and the remaining patients had no treatment except for one with a preexisting permanent atrial fibrillation receiving anticoagulant treatment.

### 3.3. Follow-Up

The follow-up duration ranged from 1 to 49 months, mean 12.7 ± 11.0 months, median 11 months. One month after implantation, full occlusion was noticed in 15 additional patients, totaling 63% of the whole population, with the remaining 16 patients having a tiny persistent shunt. One of them with a multifenestrated defect was a foreign patient who was lost to follow-up after one month when echocardiography revealed only a tiny intra-prosthetic insignificant shunt. At one-month follow-up, the mean LV end-diastolic diameter z-score decreased from +1.71 ± 1.30 to +0.28 ± 1.42 (*p* < 0.001). All patients receiving medication for heart failure could stop the treatment. Four other patients had no shunt at the six-month control, totaling a full occlusion rate of 72% at that time. Two others with residual shunts had no shunt at the 12-month control (full occlusion in 77% of the whole population). No AV blocks were noticed and hemolysis on blood test controls was not observed in patients with persistent shunts. Two patients had a grade II tricuspid regurgitation decreasing to grade I, the tricuspid insufficiency resolved in three patients, and two others had apparition of a tiny grade I tricuspid regurgitation after device occlusion. Similarly, no progression or new onset of an aortic regurgitation was observed in all patients: one patient had a grade II aortic insufficiency decreasing to grade I, and four no longer had aortic regurgitation. Of the 15 patients with persistent small aortic regurgitation, 2 of them also had a bicuspid valve. The three patients with infundibular stenosis had no change in the systolic gradient 6 months after device implantation but both systolic gradients were minimal and did not exceed 20 mm Hg. One of them with an associated mild subaortic membrane had no progression of the stenosis at the 6-month control (peak systolic gradient of 14 mm Hg).

Among the 13 patients with multiple right-sided exits, 7 had no shunt during follow-up (Figure 2 and Figure 3). Among the five patients with both multifenestrated defects and Gerbode-type VSD, three achieved complete occlusion and the other two had tiny residual shunts. All the residual shunts were trivial shunts persistent at 1 and 6 months of follow-up.

## 4. Discussion

The MFO device appears effective and safe for the transcatheter closure of PmVSD with a high rate of procedural success. Due to the plug softness and its design on the left side, the occluder fits very well within the aneurysm reducing the risk of residual shunt. It also seems really appropriate in cases of multiple exits on the right side or multifenestrated defects including the Gerbode-type VSD [1,3,4,5,6,7,8,9,10], which is not a limitation although only one exit is crossed by the device. In this setting, to obtain full occlusion was more complex but possible, and this was noticed in half of our patients, even if two patients received a second occluder, as already reported [9]. The remaining ones had only a small-to-trivial residual shunt which was hemodynamically insignificant. In fact, both the whole plug on the left side covering the LV entry and the retention disk on the right side could achieve full occlusion of the defect. Haddad has suggested that device undersizing may be associated with a persistent shunt in patients with a multifenestrated defect and should be avoided [3,10]. In addition, the possibility of connecting the device to the delivery cable from either the left or the right side is also a real advantage. It facilitates the procedure and reduces the radiation dose at implantation especially when the left-sided connection is used, thus making the creation of a complete arteriovenous loop unnecessary. Kuswiyanto has clearly shown that the fluoroscopy time was significantly reduced using the retrograde approach: 18.1 min versus 23.7 min using the antegrade approach [5]. In the present study, the mean fluoroscopy time was clearly much lower at 6.13 min using the retrograde approach—much lower than ever reported [3,7]—and 19.2 min using the antegrade approach. Thus, the retrograde approach should be clearly preferred and can simplify the procedure [1]. However, the antegrade approach could remain of interest for use in small children with large defects, in cases of a deficient aortic rim, or when there is a trivial aortic regurgitation or a mild aortic valve prolapse [5].

One of the main limitations of this technique is the risk of valvular lesion involving both the tricuspid and/or the aortic valves. A minimum distance between the defect and the aortic valve has been classically recommended on the left side especially in non-aneurysmal defects (2–3 mm) [2,5]. In contrast, device placement within a true aneurysm can minimize the risk of aortic regurgitation by locating the occluder farther from the aortic valve [2,3]. However, an aortic valve regurgitation is not rare in combination with a PmVSD. In some patients, this is related to a true aortic valve prolapse favored by the subaortic VSD and hemodynamic conditions due to the well-known Venturi effect. Aortic regurgitation was observed in 19 patients in our population and was related to a mild aortic valve prolapse in only 4 of them. This regurgitation did not worsen after implantation and no new onset above degree one occurred in the present work, as also reported by others [2,3,4,11,12]. With natural progression, valve prolapse and aortic regurgitation may worsen over time, leading to the decision of surgical VSD closure even if the left-to-right shunt remains minimal. Additionally, when the surgical approach is used, PmVSD closure alone is frequently performed when the aortic regurgitation is mild. It seems that the choice to close the PmVSD may be worthwhile in some patients to avoid the progression of the aortic valvular prolapse by suppression of the hemodynamic conditions, and this should be questioned in the future as another possible indication for percutaneous VSD closure [12].

On the other hand, tricuspid regurgitation can be observed after implantation by protrusion of the right-sided disc against the valve/chordae, and repositioning of the device is usually possible [7]. However, it seems that the retrograde approach may also reduce such risk by avoiding or limiting contact with the tricuspid valve apparatus [2], which is probably an additional benefit of this approach. Here, we would like to emphasize the fact that a complete assessment of both the tricuspid and aortic valves is necessary before device positioning on TEE to rule out any valvular regurgitation and its mechanism. Such an evaluation will allow a comparison in case of a new aortic and/or tricuspid regurgitation appearing after device placement and will help with the decision of a repositioning or release.

The use of anti-platelet therapy is frequently employed in various publications. Here, only four patients received aspirin after device release during the early experience. In fact, most patients had no treatment, as in transcatheter closure of the arterial duct. It seems that this approach is feasible and does not carry more risk for the patients in terms of thrombo-embolic events but this will need to be confirmed by other studies.

PmVSD closure carries many drawbacks. Multiple conduction abnormalities have been reported and the risk of a complete AV block is the major limitation of this technique. It has been observed with all the devices, including the Konar-MFO, and its incidence is around 1.1% according to the meta-analysis by Santhanan [13] and the publication by Tanidir [2]. Different mechanisms have been advocated to explain conduction disturbance. For the occurrence of an early AV block after closure, a mechanical trauma from the device itself or the wire/long delivery sheath has been hypothesized. Tissue lesions from compression, fibrosis, or inflammation could be the trigger of the late block [2]. In the present study, no complete AV block was observed but all these patients require a close follow-up because late block has also been reported a few years after implantation. However, the particular flexibility of the Lifetech Konar-MFO seems to reduce the risk of AV block [7]. In a multicentric study [2], a complete AV block was noticed in one patient (1%) and normal AV conduction was restored after surgical device removal and VSD closure. In this setting, the use of steroids, isoprenaline, and permanent pacemaker implantation have also been reported.

The incidence of residual shunting is quite low and will decrease with the duration of follow-up and the endothelialization of the device. Immediate full occlusion was noticed in 28% of patients in our experience and the closure rate rose to 63%, 72%, and 77% at 1 month, 6 months, and 1 year, respectively. These late results are not very different from those reported in the literature [3,7] such as the 81% closure rate at 6 months published by others [7,14]. They could be explained by the median follow-up of only 11 months in the present study. Among these patients, the residual shunt was not associated with any hemolysis, which was always checked. It should be added that all persistent shunts were trivial to small intra-prosthetic and hemodynamically insignificant shunts as shown by the decrease in LV dimension. It seems that most cases do not require any further intervention [2].

The other complication is the risk of device embolization towards either the right-sided chambers, pulmonary arteries, or the aorta. This results mainly from underestimation of the defect size but has also been related to the classical learning curve [2]. Retrieval of the embolized device can be achieved using a lasso catheter and implantation of a larger device is sometimes performed [2,3,7]. In the present study, the change to a larger device was performed once but no embolization was observed. Hemolysis is another drawback due to a residual shunt with red cell destruction by high velocity jet against the occluder. This was not noticed in the present study despite the existence of a small residual shunt, and was probably more frequent with the use of coils in our experience [14]. Sometimes hemolysis will disappear by itself with full occlusion of the defect. Otherwise, transfusion, surgical removal, or implantation of a second device to occlude the residual leak can be therapeutic options. At last, as is well known with PmVSD, there is a theoretical risk of endocarditis with this device, which is fortunately not yet reported. Preventive measures such as antibioprophylaxis are usually recommended for a period of 6 months after implantation and should be prolonged as long as a persistent shunt is observed.

This study carries some limitations. First, it is a retrospective single-center study with a limited number of patients that may not cover all the anatomical aspects of this disease. Second, outcomes and results may be influenced by the interventionist’s learning curve. Third, the duration of follow-up is relatively short and may underestimate the risk of complete late AV block. 

## 5. Conclusions

The Konar-MFO was successfully and safely used to close PmVSDs. Its design offers multiple advantages and a high rate of occlusion even in patients with multiple right-sided exits or associated mild aortic regurgitation. It appears that the retrograde approach is convenient and as effective as the antegrade approach with a significant reduction in radiation dose; therefore, it should be recommended. Multifenestrated defects, including the Gerbode VSDs, are eligible for transcatheter closure with good results. More patients are definitely needed to establish the safety and efficacy of the Konar-MFO in long-term follow-up.

## Figures and Tables

**Figure 1 jcm-12-06370-f001:**
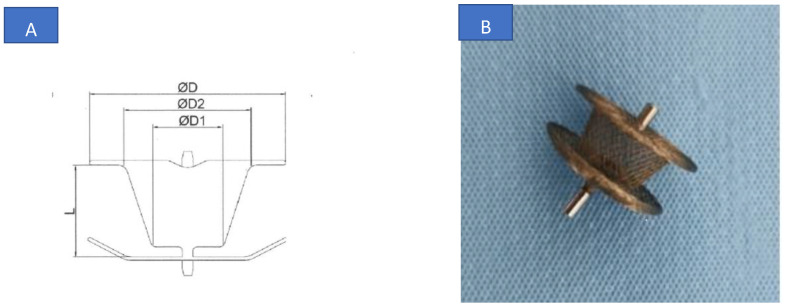
(**A**) Konar-MFO made of 2 discs joined by a soft articulation with two connecting micro-screw systems attached on the external surfaces of both the right and the left discs. (**B**) Device sizes are categorized according to D2 waist diameter (LV size) or D1 waist diameter (RV size). (**B**) Color picture.

**Figure 2 jcm-12-06370-f002:**
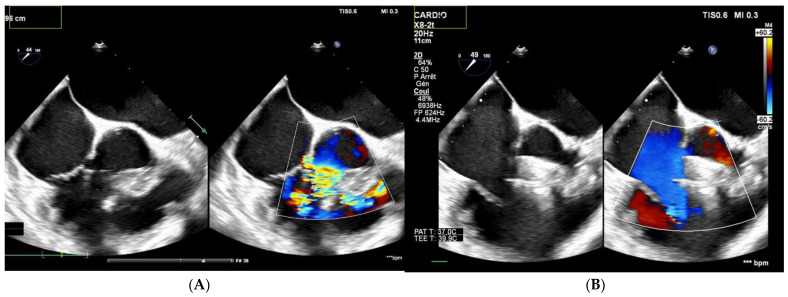
Patient with multifenestrated PmVSD. (**A**) Left: transesophageal echocardiography, notice the 2 jets to the RV. (**B**) After closure with MFO 14/12 mm device, no shunt is observed on color Doppler echocardiography.

**Figure 3 jcm-12-06370-f003:**
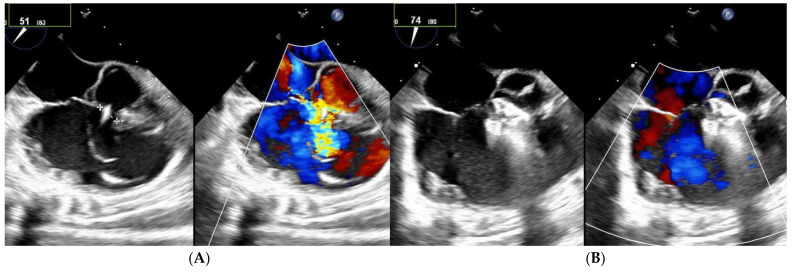
Patient with multifenestrated PmVSD and Gerbode-type VSD. (**A**) Notice the large aneurysm with a basis of 9 mm on left side. (**B**) After implantation of 8–10 mm Konar device, no residual shunt is observed.

## Data Availability

Not applicable.

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
