# Peer review of "Transcatheter Closure of Perimembranous Ventricular Septal Defects Including Multifenestrated and Gerbode-Type Defects Using the Lifetech Konar Device"

_jcm, 2023, doi:10.3390/jcm12196370_

Round 1

Reviewer 1 Report

Percutaneous closure of ventricular septal defects remains a contemptuous topic in North America since surgical results are generally excellent and the prior reports of complete heart block have deterred people from using this approach frequently. As such, seeing reports of successful transcatheter perimembranous VSD closures is encouraging and important. A few comments:

1) In the Study population section - were there any patients that were referred but excluded or declined transcatheter closure due to certain anatomic characteristics such as size of defect? 

2) In the Results section - the LV mean Z score is reported to be 1.7 (dilated is > +2). Could the authors describe what the indication of closure was in patients with EDD < +2?

3) Since the title of the work particularly mentions defects with multiple fenestrations and Gerbode defects, a description of how the procedure differed in these patients (where was the wire crossed from, device selection) would be helpful

4) Under implantation section - could the authors describe if there was 24-hour continuous ECG/telemtry monitoring during the post procedure admission and if any arrhythmias were noted during this admission? 

5) The results on patients with multiple fenestrations need to be clarified a little. There were 5 patients with Gerbode type VSD with 3/5 achieving complete resolution of shunt and the other 2 with a tiny residual shunt? If that is correct, it should be clearly stated since that seems to be the novel aspect of this study compared to the prior works by Tanidir and Santhanam et al.

6) The percentage of patients with residual shunts is a lot higher in this study compared to prior works reporting 8-16% - the author attribute this to the short follow up period. Could the device size selection have anything to do with this?

Finally, in the description the authors discuss how the indication for pmVSD closure may change given the success of the trans-catheter approach. These findings are not supported by this study and could likely be removed. 

Minor edits needed for spelling/punctuation. 

Reviewer 2 Report

This is a well written single centre retrospective report on the techniques and results of transcatheter perimembraneous VSD closure using a specific device. The methods are sound, and the results add to the growing evidence that catheter interventional defect closure is a safe and effective alternative to surgery.

Despite the novelty being limited, there are already even more studies available than the authors already cited: e.g., Yildiz K, et al. Eur Rev Med Pharmacol Sci. 2023 May;27(9):4053-4059 (22 children < 10 kg) and Haddad RN, Saliba ZS. Front Cardiovasc Med. 2023 Jul 5; 10:1215397. eCollection 2023 (33 children).

The authors state that all patients enrolled not only had shunt-enlarged left ventricles but were symptomatic from exertional dyspnoea, how was this ascertained?

The implantation techniques used by the authors seem not to be device-specific and are common knowledge so this section could be shortened to the device specific details.

There is one more aspect the reviewer is not able to understand fully: The authors are part of a nationwide group conducting a prospective study called FRANCISCO and coauthors of the published study protocol (Guirgis L, et al. Cardiol Young. 2021 Oct;31(10):1557-1562). So at least a subgroup of their patients should be enrolled there. The authors should mention this prospective study and comment on its impact on patient selection. Was there double enrolment? 

Round 2

Reviewer 2 Report

This is a well written single centre retrospective report on the techniques and results of transcatheter perimembraneous VSD closure using a specific device. The methods are sound, and the results add to the growing evidence that catheter interventional defect closure is a safe and effective alternative to surgery. Therefore the report should be published as is.